# Nearest-Neighbour and Non-Nearest-Neighbour Non-Covalent Interactions between Substituents in the Aromatic Systems: Experimental and Theoretical Investigation of Functionally Substituted Benzophenones

**DOI:** 10.3390/molecules27238477

**Published:** 2022-12-02

**Authors:** Artemiy A. Samarov, Stanislav O. Kondratev, Sergey P. Verevkin

**Affiliations:** 1Department of Chemical Thermodynamics and Kinetics, Saint Petersburg State University, 198504 Saint Petersburg, Russia; 2Competence Centre CALOR, Department of Physical Chemistry, Faculty of Interdisciplinary Research, University of Rostock, 18059 Rostock, Germany; 3Department of Physical Chemistry, Kazan Federal University, 420008 Kazan, Russia

**Keywords:** vapour pressure, enthalpy of vaporisation, enthalpy of formation, quantum chemical calculations, group-additivity, structure–property relationships

## Abstract

Benzophenone derivatives exhibit not only biological activity but also act as photo initiator and UV blocker. We carried out experimental and theoretical thermochemical studies of hydroxy- and methoxy-substituted benzophenones. Standard molar enthalpies of vaporisation were obtained from the temperature dependence of vapour pressures measured by the transpiration method. The thermodynamic data on phase transitions available in the literature (crystal–gas, crystal–liquid, and liquid–gas) were also collected and evaluated. High-level quantum chemical methods G3MP2 and G4 were used to estimate the standard molar enthalpies of formation of substituted benzophenones in the gas phase and establish agreement between experimental and theoretical results. The application of the “centrepiece” group-contribution approach to hydroxy- and methoxy-substituted benzophenones was demonstrated. A quantitative assessment of the hydrogen bond was carried out using various approaches based on experimental data and quantum chemical calculations.

## 1. Introduction

Hydroxy- and methoxy-benzophenones represent an important class of biologically active compounds. They exhibit important pharmaceutical properties such as cytotoxic effects against various cancer cells [1]. The attractive features of these compounds are commercial availability and their UV absorbing power. The benzophenones are used as cosmetics and medicine for their ability to harmlessly absorb and scatter UV radiation, protecting products and human skin from the harmful effects of UV radiation. One of the most important areas of application for benzophenones is polymers. The polymers are often exposed to ultraviolet radiation and the absorption of this energy causes the macromolecular chains to break. In order to slow down this process for outdoor use, these materials must be protected with an appropriate UV absorber, e.g., a suitable benzophenone. In order to select an appropriate stabiliser, it is necessary to evaluate the efficiency of a particular compound in terms of the potential for loss from the polymer through evaporation. The evaporation rate is controlled by the vapor pressure of the compound, so it is important to know this parameter at any temperature of application [2]. The benzophenones are low-volatile compounds and accurate vapour pressure measurement is a challenging task. In this work, the experimental focus was on the investigation of the phase transition thermodynamics of hydroxy- and methoxy-substituted benzophenones.

Here we present results on vapour pressures, enthalpies of phase transitions, and enthalpies of formation of a series of substituted benzophenones of general formulas given in Figure 1 and Figure 2. The data available in the literature and new experimental results were evaluated and checked for internal consistency. The consistent thermochemical data sets for substituted benzophenones were used for the design and development of a “centrepiece” group-contribution approach necessary for assessing the enthalpies of vaporisation and enthalpies of formation of compounds that are important for process engineering calculations.

In the course of the thermochemical measurements, another aspect of the properties of benzophenones attracted our attention. This aspect is not related to our research field but has inspired a deeper interpretation of our thermochemical results. It was found [3] that by absorbing the UV light, benzophenones transform into an excited state, but harmlessly dissipate the absorbed energy and return to the ground state, i.e., they convert the absorbed photons to heat without being chemically affected. The UV stabilization process most likely occurs due to the scavenging of free radicals [4,5]. It has turned out that a mechanism of this process involves intra-molecular hydrogen bonding (intra-HB), specific for ortho-substituted benzophenones since the radical product is no longer capable of forming the intra-HB [5]. Most benzophenones shown in Figure 1 and Figure 2 bear one or two intra-HB. Qualitative evidence for the presence of these intra-HBs in 2-hydroxy-substituted benzophenones can be found in sufficient detail in the literature. However, quantitative data on the strength of the intra-HB in such molecules is limited [6]. Therefore, we used the thermochemical datasets evaluated in this work for the proper quantification of the intra-HB strength using thermodynamic and quantum chemical methods.

## 2. Materials and Methods

Commercially available samples of substituted benzophenones were used in this work: 2,2′-dihydroxy-benzophenone (TCI, Eschborn, Germany, ˃99%), 2,4-dihydroxy-benzophenone (TCI, Germany, ˃99%), 2-hydroxy-4-methoxy-benzophenone (TCI, Germany, ˃99%), 2,2′-hydroxy-4-methoxy-benzophenone (TCI, Germany, ˃98%), 2,2′,4,4′-tetrahydroxy-benzophenone (TCI, Germany, ˃98+%), and 2,2′-dihydroxy-4,4′-dimethoxy-benzophenone (Sigma-Aldrich, Darmstadt, Germany, ˃98%). Solid samples were purified by the fractional sublimation in a vacuum. Purities were determined using a gas chromatograph equipped with a capillary column HP-5 and a flame ionization detector. No impurities (greater than 0.005 mass fraction) were detected in samples used for vapour pressure measurements. Vapour pressures of substituted benzophenones at different temperatures were measured by using the transpiration method [7,8] and Knudsen effusion method [9]. The standard molar enthalpies of vaporisation/sublimation, $\Delta_{l,\mathrm{cr}}^{g}H_{m}^{o}$, were derived from the temperature dependences of vapour pressures. Melting temperatures and enthalpies of fusion were measured by DSC [10]. The quantum chemical composite method G4 [11] from Gaussian 16 software (Revision A.03, Wallingford, CT, USA) [12] was used for calculations of enthalpies *H*_298_-values, which were finally converted to the gas-phase standard molar enthalpies of formation $\Delta_{f}H_{m}^{o}$(g) and discussed. Brief descriptions of the experimental techniques used in this work are included in the Appendix A.

## 3. Results and Discussion

### 3.1. Absolute Vapour Pressures and Thermodynamics of Vaporisation/Sublimation

The vapour pressures, *p_i_*, for the substituted benzophenones and their temperature dependences measured by the transpiration method are given in Table 1.

The vapour pressures, *p_i_*, for 2,4-di-hydroxy-benzophenone and 2,2′4,4′-tetra-hydroxy-benzophenone were too low to be measured within a reasonable time using the transpiration method. Their temperature dependences were measured using the Knudsen effusion method. The results of the Knudsen method are given in Table 2.

Vapour pressure results for the substituted benzophenones were fit to the following equation [7,8]:(1)$$R\times \ln(p_{i}/p_{ref})=a+\frac{b}{T}+\Delta_{l,\mathrm{cr}}^{g}C_{p,m}^{o}\times \ln\left(\frac{T}{T_{0}}\right)$$
where R = 8.31446 J·K^−1^·mol^−1^ is the molar gas constant, the reference pressure, $p_{ref}=1\;\mathrm{Pa},$ and a and b are adjustable parameters; the arbitrary temperature *T*_0_ applied in Equation (1) was chosen to be *T*_0_ = 298.15 K and $\Delta_{l,\mathrm{cr}}^{g}C_{p,m}^{o}$ is the difference of the molar heat capacities of the gas and the liquid/crystal phases, respectively. The isobaric heat capacity differences $\Delta_{l,\mathrm{cr}}^{g}C_{p,m}^{o}$ required for temperature adjustments of vaporisation/sublimation enthalpies are given in Table 3.

The vapour pressures at different temperatures, *T*, measured in this work, as well as those available from the literature, were used to derive the enthalpies of sublimation/vaporisation using the following equation:(2)$$\Delta_{\mathrm{cr},l}^{g}H_{m}^{o}\left(T\right)=-b+\Delta_{\mathrm{cr},l}^{g}C_{p,m}^{o}\times T$$

Sublimation entropies at temperatures *T* were also derived from the vapour pressures temperature dependences using Equation (3):(3)$$\Delta_{\mathrm{cr},l}^{g}S_{m}^{o}\left(T\right)=\Delta_{\mathrm{cr},l}^{g}H_{m}^{o}/T+R\ln\left(p_{i}/p^{o}\right)$$
with $p^{o}$ = 0.1 MPa. Experimental vapour pressures measured at different temperatures, coefficients a and b of Equation (1), as well as values of $\Delta_{\mathrm{cr},l}^{g}H_{m}^{o}$(*T*) and $\Delta_{\mathrm{cr},l}^{g}S_{m}^{o}$(*T*) are given in Table 1 and Table 2. The method for calculating the combined uncertainties of the vaporisation/sublimation enthalpies involves uncertainties from the experimental conditions of transpiration, uncertainties from the vapour pressure, and uncertainties due to the temperature adjustment to *T* = 298.15 K, as described elsewhere [13,14]. The compilation of available standard molar vaporisation/sublimation enthalpies for the compounds shown in Figure 1 and Figure 2 is given in Table 4. The original absolute vapour pressures available in the literature were also treated using Equations (2) and (3) to evaluate the enthalpies of vaporisation/sublimation at 298.15 K (see Table 4) in the same way as our own results. The uncertainties of the literature results were also re-assessed in the same way [13,14], as for our own experimental results.

The $\Delta_{\mathrm{cr},l}^{g}C_{p,m}^{o}$-values used in Equations (1) and (2) are usually derived from empirical equations developed by Chickos and Acree [19,20]:(4)$$\Delta_{\mathrm{cr}}^{g}C_{p,m}^{o}\;\left(298.15\;K\right)=-0.15\times C_{p,m}^{o}\;\left(\mathrm{cr},\;298.15\;K\right)-0.75$$
(5)$$\Delta_{l}^{g}C_{p,m}^{o}\;\left(298.15\;K\right)=-0.26\times C_{p,m}^{o}\;\left(\mathrm{liq},\;298.15\;K\right)-10.58$$
where $C_{p,m}^{o}$(cr, 298.15 K) or $C_{p,m}^{o}$(liq, 298.15 K) values are of the experimental origin or can also be estimated using the group-additivity method [17].

The vaporisation/sublimation at 298.15 K measured in this work (see Table 1 and Table 2) and those available in the literature and derived using Equation (2) are compiled in Table 4.

**Table 4 molecules-27-08477-t004:** Compilation of enthalpies of vaporisation/sublimation $\Delta_{\mathrm{cr},l}^{g}H_{m}^{o}$ for the benzophenone derivatives derived in this work and from the data available in the literature.

Compound	M ^a^	*T*-Range	$\Delta_{l,\mathrm{cr}}^{g}H_{m}^{o}(T_{\mathrm{av}})$	$\Delta_{l,\mathrm{cr}}^{g}H_{m}^{o}\;(298\;K){\;}^{b}$	Ref.
CAS		K	kJ·mol^−1^	kJ·mol^−1^	
4-methoxy-benzophenone (liq) [611-94-9]	*J_x_*			93.5 ± 2.0	Table 5
	*T_b_*	628.2 [21]		90.2 ± 2.0	Table 6
				**91.8 ± 1.4** ^d^	average
benzophenone (liq)				77.6 ± 0.5	Table 7
2-hydroxy-benzophenone (cr) [117-99-7]	C	401.6	123.1 ± 1.7	97.9 ± 1.9	[15]
2-hydroxy-benzophenone (liq)	*PhT*			80.2 ± 2.0	Table 7
3-hydroxy-benzophenone (cr) [13020-57-0]	K-QCM	361.2–378.2	129.9 ± 0.7	131.7 ± 0.9	[15]
3-hydroxy-benzophenone (liq)	*PhT*			110.8 ± 2.2	Table 7
4-hydroxy-benzophenone(cr) [1137-42-4]	K-QCM	377.2–394.2	128.6 ± 0.7	130.3 ± 1.0	[15]
4-hydroxy-benzophenone (liq)	*PhT*			113.7 ± 2.5	Table 7
2-methoxy-benzophenone (liq) [2553-04-0]				87.4 ± 3.0 ^c^	this work
3-methoxy-benzophenone (liq) [6136-67-0]	*T_b_*	621.4 [21]		88.7 ± 2.0	Table 6
2,4-di-hydroxy-benzophenone (cr)	IG	323–363	93 ± 16	(95 ± 16)	[22]
[131-56-6]	n/a	312–353	134.0 ± 2.0	135.3 ± 2.1	[23]
	K	366.6–406.8	129.5 ± 2.8	133.0 ± 2.9	Table 2
				**134.5 ± 1.7** ^d^	average
2,4-di-hydroxy-benzophenone (liq)	UVS	418–485	86.8 ± 5.0	(105.8 ± 5.0)	[24]
	*PhT*			112.3 ± 4.2	Table 7
2-hydroxy-4-methoxy-benzophenone (cr)	n/a	281–337	118.8 ± 5.0	119.3 ± 5.0	[23]
[131-57-7]	K	306.5–320.5	109.0 ± 5.0	109.6 ± 5.1	[25]
	*PhT*			117.1 ± 1.3	Table 7
				**116.8 ± 1.2** ^d^	average
2-hydroxy-4-methoxy-benzophenone (liq)	UVS	337–413	74.2 ± 5.0	(83.5 ± 5.0)	[24]
	T	341.2–368.2	91.6 ± 0.9	98.4 ± 1.0	Table 1
2,2′-dihydroxy-benzophenone (cr) [835-11-0]	*PhT*			101.9 ± 1.3	Table 7
2,2′-dihydroxy-benzophenone (liq)	T	363.1–408.4	74.7 ± 0.8	84.9 ± 0.9	Table 1
2,2′-dihydroxy-4-methoxy-benzophenone (cr)	n/a	303-342	228 ± 5	(229 ± 5.0)	[23]
[131-53-3]	*PhT*			118.6 ± 1.6	Table 7
2,2′-dihydroxy-4-methoxy-benzophenone (liq)	TGA	343–573	81.8 ± 0.2	103.9 ± 4.4	[26]
[131-53-3]	UVS	342–481	74.1 ± 5.0	(89.7 ± 5.0)	[24]
	T	365.1–430.9	87.1 ± 0.7	100.7 ± 0.9	Table 1
				**100.8 ± 0.9** ^d^	average
2,2′,4,4′-tetrahydroxy-benzophenone (cr)	n/a	363–471	143.2 ± 5.0	(148.7 ± 5.0)	[23]
[131-55-5]	K	431.8–453.8	153.6 ± 2.6	159.8 ± 3.1	Table 2
2,2′,4,4′-tetrahydroxy-benzophenone (liq)	TGA	472–573	150.5 ± 0.2	(186.2 ± 7.1)	[26]
	*PhT*			152.0 ± 6.8	Table 7
2,2′-dihydroxy-4,4′-dimethoxybenzophenone (cr)	n/a	325–408	146.8 ± 5.0	(150.4 ± 5.0)	[23]
[131-54-4]	T	368.5–399.3	134.8 ± 2.2	139.4 ± 2.4	Table 1
2,2′-dihydroxy-4,4′-	TGA	412–573	96.9 ± 0.2	126.7 ± 6.0	[26]
dimethoxybenzophenone (liq)	K	412.6–435.1	98.0 ± 2.8	117.2 ± 2.9	Table 1
	*PhT*			117.6 ± 4.2	Table 7
				**118.6 ± 2.2** ^d^	average

^a^ Techniques: T = transpiration method; C = Calvet microcalorimetry; K = Knudsen effusion method; K-QCM = Knudsen effusion method combined with the quartz-crystal microbalance for the mass loss measurements; UVS = ultraviolet spectrophotometry; IG = ionisation gauge; n/a = not available; *J_x_*—from correlation of experimental vaporisation enthalpies with Kovats’s indices (see text); S = static method; *PhT* = calculated according to Equation (8) (see text); TGA = thermogravimetry. ^b^ Vapour pressures available in the literature were treated using Equations (2) and (3) with help of heat capacity differences from Table 3 to evaluate the enthalpy of vaporisation/sublimation at 298.15 K in the same way as our own results in Table 1 and Table 2. Uncertainty of the vaporisation/sublimation enthalpy *U*($\Delta_{l,\mathrm{cr}}^{g}H_{m}^{o}$) is the expanded uncertainty (0.95 level of confidence) calculated according to procedure described elsewhere [13,14]. ^c^ Assessed based on vaporisation enthalpy of 3-methoxy-benzophenone (this table) and the difference between vaporisation enthalpy of 2-methoxy- and 3-methoxy-acetophenone [27]. ^d^ Weighted mean value. Values in parenthesis were excluded from the calculation of the mean. Values in bold are recommended for further thermochemical calculations.

The data on vaporisation/sublimation enthalpies compiled In Table 4 are of different quality. Schmitt and Hirt [24] studied the volatility of benzophenones by measuring their rate of evaporation in a vacuum from a free surface which was monitored spectrophotometrically (UVS method). The latter method is not commonly used for vapor pressure measurements. Most likely, this method was not sufficiently developed for the measurement of low pressures and for this reason, the temperature dependencies of the vapour pressure showed an incorrect slope and the significantly lower vaporisation enthalpies compared to other established methods. Price and Hawkins [26] used a thermogravimetric method to measure the vapour pressures of a series of hydroxy-benzophenones. Enthalpies of vaporisation determined from the slope of a plot of the logarithm of the vapour pressure against reciprocal absolute temperature agreed reasonably well (see 2,2′-dihydroxy-4-methoxy-benzophenone and 2,2′-dihydroxy-4,4′-dimethoxy-benzophenone in Table 4) with results from other methods (except for 2,2′,4,4′-tetra-hydroxy-benzophenone). Only single experimental results were found in the literature for the hydroxy-benzophenones. However, the sublimation enthalpies for these compounds were measured using the established methods and in the laboratory with very good experience [15] and are considered to be reliable.

### 3.2. Consistency of Results on Vaporisation/Sublimation Enthalpies

#### 3.2.1. Kovats Retention Indices for Validation of Experimental Vaporisation Enthalpies

In the situation of insufficient experimental data of the $\Delta_{l}^{g}H_{m}^{o}$ (298.15 K)-values or their inconsistency, correlation with chromatographic retention indices available for substituted benzophenones [28] can be a useful tool for verification of available data [29]. It is established that the $\Delta_{l}^{g}H_{m}^{o}$ (298.15 K)-values correlate linearly with Kovats indices [30] in various homologous series of alkanes, alkylbenzenes, aliphatic ethers, alcohols, or in a series of structurally similar compounds [29]. We apply this approach to a set of molecules (see Table 5), where *J_x_*–values and the $\Delta_{l}^{g}H_{m}^{o}$ (298.15 K)-values were available from the literature. 

**Table 5 molecules-27-08477-t005:** Correlation of vaporisation enthalpies, $\Delta_{l}^{g}H_{m}^{o}$ (298.15 K), of benzene and benzophenone derivatives with their Kovats indices (*J_x_*).

	*J_x_* ^a^	$\Delta_{l}^{g}H_{m}^{o}\;(298\;K{)}_{\exp}$	$\Delta_{l}^{g}H_{m}^{o}\;(298\;K){{\mathrm{calc}}_{\;}}^{b}$	Δ ^c^
Compound		kJ·mol^−1^	kJ·mol^−1^	kJ·mol^−1^
methoxybenzene	900	46.6 ± 0.2 [31]	46.8	−0.2
acetophenone	1048	55.4 ± 0.3 [32]	54.5	0.9
2′-methoxy-acetophenone	1269	64.6 ± 0.4 [27]	65.9	−1.3
3′-methoxy-acetophenone	1279	65.8 ± 0.4 [27]	66.4	−0.6
4′-methoxy-acetophenone	1327	70.4 ± 0.6 [27]	68.9	1.5
**4**′**-methoxy**-**benzophenone**	1804 [33]	-	**93.5 ± 2.0**	

^a^ Kovats indices, *J_x_*, on the standard non-polar column SE-30 [34]. ^b^ Calculated using equation: $\Delta_{l}^{g}H_{m}^{o}$ (298.15 K)/(kJ·mol^−1^) = 0.0516 × *J_x_* + 0.4 with (*R*^2^ = 0.9942) with the assessed uncertainty of ±2.0 kJ·mol^−1^ (expanded uncertainty 0.95 level of confidence). ^c^ Difference between column 3 and 4 in this table.

The following linear correlation was obtained when the $\Delta_{l}^{g}H_{m}^{o}$ (298.15 K)-values were correlated with *J_x_*-values for this set of compounds:(6)$$\Delta_{l}^{g}H_{m}^{o}\;(298.15\;K)/(\mathrm{kJ}\cdot {\mathrm{mol}}^{-1})=0.0516\;\times \;J_{x}+0.4\;\mathrm{with}\;(R^{2}=0.9942)$$

The relationship according to Equation (6) allowed estimation of the unknown vaporisation enthalpy for 4′-methoxy-benzophenone (see Table 5). This “theoretical” result, derived from correlation with the Kovats indices, is denoted *J_x_* and is used in Table 4 for comparison with another result for this compound obtained from correlation with normal boiling temperatures, as shown in the next section.

#### 3.2.2. Normal Boiling Temperatures for Validation of Experimental Vaporisation Enthalpies

Another way to determine the consistency of the experimental results on vaporisation enthalpies for substituted benzophenones is also to correlate vaporisation enthalpies with normal boiling temperatures [35]. For a set of structurally similar compounds (see Table 6) with well-established $\Delta_{l}^{g}H_{m}^{o}$ (298.15 K)-values and known normal boiling temperatures, *T_b_*, we derived the following linear correlation: (7)$$\Delta_{l}^{g}H_{m}^{o}\;(298.15\;K)/(\mathrm{kJ}\cdot {\mathrm{mol}}^{-1})=0.2184\;\times \;T_{b}-47.0\;\;\;\;\;\;\;\;\;\mathrm{with}\;(R^{2}=0.9936)$$

**Table 6 molecules-27-08477-t006:** Correlation of vaporisation enthalpies, $\Delta_{l}^{g}H_{m}^{o}$ (298.15 K), of benzene and benzophenone derivatives with their normal boiling temperatures (*T_b_*).

	*T_b_* ^a^	$\Delta_{l}^{g}H_{m}^{o}\;(298\;K{)}_{\exp}$	$\Delta_{l}^{g}H_{m}^{o}\;(298\;K{)}_{\mathrm{calc}}{\;}^{b}$	Δ ^c^
Compound		kJ·mol^−1^	kJ·mol^−1^	kJ·mol^−1^
methoxybenzene	426.8	46.6 ± 0.2 [31]	46.2	0.4
acetophenone	475.8	55.4 ± 0.3 [32]	56.9	−1.5
2′-methoxy-acetophenone	(511.5) ^d^	64.6 ± 0.4 [27]	64.6	0.0
3′-methoxy-acetophenone	513.2	65.8 ± 0.4 [27]	65.1	0.7
4′-methoxy-acetophenone	531.2	70.4 ± 0.6 [27]	69.0	1.4
benzophenone	578.6	78.0 ± 0.2 [36]	79.4	−1.4
2′-methyl-benzophenone	582.7	81.2 ± 1.7 [37]	80.3	0.9
3′-methyl-benzophenone	586.2	80.7 ± 0.4 [36]	81.0	−0.3
4′-methyl-benzophenone	(583.6) ^d^	80.5 ± 0.5 [36]	80.5	0.0
**3′-methoxy-benzophenone**	621.5 [21]		**88.7 ± 2.0**	
**4′-methoxy-benzophenone**	628.2 [21]		**90.2 ± 2.0**	

^a^ Normal boiling temperatures, *T_b_*, [23]. ^b^ Calculated using Equation (6): $\Delta_{l}^{g}H_{m}^{o}$ (298.15 K)/(kJ·mol^−1^) = 0.2184 × *T_b_* − 47.0 with (*R*^2^ = 0.9936) and with the assessed uncertainty of ±2.0 kJ·mol^−1^ (expanded uncertainty 0.95 level of confidence). ^c^ Difference between columns 3 and 4 in this table. ^d^ Assessed by Equation (6).

The relationship according to Equation (7) allowed estimation of the unknown enthalpies of vaporisation for 3′-methoxy-benzophenone and 4′-methoxy-benzophenone (see Table 6). These “theoretical” results, derived from correlation with normal boiling temperatures, are denoted in Table 4 as *T_b_*–values. As can be seen in Table 4, the vaporisation enthalpies of 4′-methoxy-benzophenone derived from the *T_b_* and *J_x_* correlations agree well within the uncertainties attributed to them. 

#### 3.2.3. Consistency of Phase Transitions Enthalpies

The common thermochemical equation:(8)$$\Delta_{l}^{g}H_{m}^{o}\;(298.15\;K)=\Delta_{\mathrm{cr}}^{g}H_{m}^{o}\;\left(298.15\;K)-\Delta_{\mathrm{cr}}^{l}H_{m}^{o}\;(298.15\;K\right)$$
can also be used to establish the consistency of experimental data on phase transitions (liquid–gas, solid–gas, and solid–liquid) measured or evaluated in this work. For example, for 2,2′-dihydroxy-4,4′-dimethoxybenzophenone, the sublimation enthalpy $\Delta_{\mathrm{cr}}^{g}H_{m}^{o}$ (298.15 K) = 139.4 ± 2.4 kJ·mol^−1^ was measured using the transpiration method below the melting point (see Table 1) and the vaporisation enthalpy $\Delta_{l}^{g}H_{m}^{o}$ (298.15 K) = 117.2 ± 2.9 kJ·mol^−1^ was derived from vapour pressures measured above the melting point (see Table 1). The consistency of phase transitions available for 2,2′-dihydroxy-4,4′-dimethoxybenzophenone can be easily established with the help of Equation (8) and the experimental enthalpy of fusion for this compound $\Delta_{\mathrm{cr}}^{l}H_{m}^{o}$ (298.15 K) = 21.8 ± 3.4 kJ·mol^−1^ (see Table 7) as follows:$$\Delta_{l}^{g}H_{m}^{o}\;(298.15\;K,\;2,2\prime -\mathrm{dihydroxy}-4,4\prime -\mathrm{dimethoxybenzophenone})=139.4-21.8=117.6\;\pm \;4.2\;\mathrm{kJ}\cdot {\mathrm{mol}}^{-1}.$$

This estimate is in excellent agreement with the transpiration experiment $\Delta_{l}^{g}H_{m}^{o}$ (298.15 K) = 117.2 ± 2.9 kJ·mol^−1^, proving the consistency of the energetics of all three phase transitions. Similarly, we used Equation (8) to derive the unknown enthalpies of vaporisation for 2-hydroxy-, 3-hydroxy-, 4-hydroxy-benzophenones, 2,4-dihydroxy-benzophenone, 2,2′-dihydroxy-4-methoxy-benzophenone, 2,2′,4,4′-tetrahydroxy-benzophenone, and 2,2′-dihydroxy-4,4′-dimethoxy-benzophenone from the known enthalpies of sublimation and enthalpies of fusion. In addition, the missing sublimation enthalpies for 2,2′-dihydroxy-benzophenone, 2-hydroxy-4-methoxy-benzophenone and 2,2′-dihydroxy-4-methoxy-benzophenone were also calculated using Equation (8) and available experimental vaporisation enthalpies as shown in Table 7.

**Table 7 molecules-27-08477-t007:** Phase transitions thermodynamics of substituted benzophenones (in kJ·mol^−1^) ^a^.

Compounds	*T*_fus_, K	$\Delta_{cr}^{l}H_{m}^{o}$ at *T*_fus_	$\Delta_{cr}^{l}H_{m}^{o}{\;}^{b}$	$\Delta_{cr}^{g}H_{m}^{o}{\;}^{c}$	$\Delta_{l}^{g}H_{m}^{o}{\;}^{d}$
				298.15 K	
1	2	3	4	5	6
benzophenone [38]	321.0 ± 0.1	18.6 ± 0.3	17.4 ± 0.5	95.0 ± 0.1	77.6 ± 0.5
2-hydroxy-benzophenone [15]	312.3 ± 0.1	18.7 ± 0.1	17.7 ± 0.3	97.9 ± 1.9	80.2 ± 2.0
3-hydroxy-benzophenone [15]	390.5 ± 0.3	27.4 ± 0.2	20.9 ± 2.0	131.7 ± 0.9	110.8 ± 2.2
4-hydroxy-benzophenone [15]	407.7 ± 0.5	24.4 ± 0.1	16.6 ± 2.3	130.3 ± 1.0	113.7 ± 2.5
2,4-dihydroxy-benzophenone [this work]	417.7 ± 0.2	30.8 ± 0.7	20.7 ± 3.1	133.0 ± 2.9	112.3 ± 4.2
2,2′-dihydroxy-benzophenone [16]	334.5 ± 0.1	20.1 ± 0.1	17.0 ± 1.0	101.9 ± 1.3 ^e^	84.9 ± 0.9 ^f^
2-hydroxy-4-methoxy-benzophenone [25]	336.7 ± 0.5	21.8 ± 0.1	18.7 ± 0.9	117.1 ± 1.3 ^e^	98.4 ± 1.0 ^f^
2,2′-dihydroxy-4-methoxy-benzophenone [26]	343.0 ± 0.5	22.0 ± 0.5	17.9 ± 1.3	118.7 ± 1.6 ^e^	100.8 ± 0.9 ^f^
2,2′,4,4′-tetrahydroxy-benzophenone [26]	472.0 ± 0.5	28.0 ± 0.5	7.8 ± 6.1	159.8 ± 3.1	152.0 ± 6.8
2,2′-dihydroxy-4,4′-dimethoxy-benzophenone [26]	412.3 ± 0.5	33.2 ± 0.5	21.8 ± 3.4	139.4 ± 2.4	117.6 ± 4.2

^a^ Uncertainties are presented as expanded uncertainties (0.95 level of confidence with k = 2). ^b^ The experimental enthalpies of fusion $\Delta_{\mathrm{cr}}^{l}H_{m}^{o}$ measured at *T*_fus_ were adjusted to *T* = 298.15 K with help of the following equation [18]: $\Delta_{\mathrm{cr}}^{l}H_{m}^{o}$ (298.15 K)/(J·mol^−1^) = $\Delta_{\mathrm{cr}}^{l}H_{m}^{o}$ (*T*_fus_/K) − ($\Delta_{\mathrm{cr}}^{g}C_{p,m}^{o}$−$\Delta_{l}^{g}C_{p,m}^{o}$) × [(*T*_fus_/K) − 298.15 K], where $\Delta_{\mathrm{cr}}^{g}C_{p,m}^{o}$ and $\Delta_{l}^{g}C_{p,m}^{o}$ were taken from Table 3. Uncertainties in the temperature adjustment of fusion enthalpies from *T*_fus_ to the reference temperature are estimated to account with 30% to the total adjustment [18]. ^c^ Experimental values of sublimation enthalpies (see Table 4). ^d^ Calculated as the difference between columns 5 and 4 in this table. ^e^ Calculated as the sum columns 6 and 4 in this table. ^f^ Experimental values of vaporisation enthalpies (see Table 4).

From Table 4, it can be seen now that for many compounds agreement among $\Delta_{l,\mathrm{cr}}^{g}H_{m}^{o}$ (298.15 K)-values, which were derived in different ways, all lie within the assigned error bars. To get more confidence and reliability, we calculated the weighted average (the uncertainty was used as a weighing factor) for the substituted benzophenones given in Table 4. These values are highlighted in bold and are recommended for thermochemical calculations.

### 3.3. Gas-Phase Standard Molar Enthalpies of Formation

Information on the enthalpies of formation of substituted benzophenones is limited in the literature. The recent development of high-level quantum chemistry methods, especially composite methods, makes it promising to calculate enthalpies of formation $\Delta_{f}H_{m}^{o}$(g) at the level of “chemical accuracy” [39,40].

This trend has made the G-family of composite methods a valuable tool for mutual validation of experimental and computational thermochemistry. A discrepancy or agreement between the *experimental* and *theoretical* $\Delta_{f}H_{m}^{o}$ (g, 298.15 K)-values could provide a criterion for mutual validation of both results. In addition, this valuable information helps in evaluating the quality of the thermochemical data for compounds under study. In order to compensate for the lack of enthalpy data and to validate the present results, the gas phase formation enthalpies for substituted benzophenones were estimated using the quantum chemical method G4 [11]. The experimental data on the $\Delta_{f}H_{m}^{o}$ (298.15 K)-values available in the literature and results of calculations are presented in Table 8.

Stable conformers were found by using a computer code named CREST (conformer-rotamer ensemble sampling tool) [43] and optimised with the B3LYP/6-31g(d,p) method [44]. The energies *E*_0_ and the enthalpies *H*_298_ of the most stable conformers were finally calculated by using the G4 method (see Figure 3 and Appendix A). 

The conformations of 3-hydroxy-, 4-hydroxy-benzophenones, as well as of 2-methoxy-, 3-methoxy, and 4-methoxy-benzophenones are generally flat and substituents are in-plane to the backbone of the benzophenone ring (see Appendix A). The conformations of 2-hydroxy substituted benzophenones are more diverse and interesting since all these molecules feature the intra-molecular hydrogen bonding (intra-HB) between the carbonyl group and the hydroxyl substituent. The most stable conformations of 2-hydroxy substituted benzophenones are given in Figure 3 (left). To assess the strength of the intra-HB as an energy difference between the hydrogen-bonded and the open form (OH group rotated 180° around the C–O axis), we also calculated the open conformers (see Figure 3, right). The discussion of these differences can be found in Section 4.

The *H*_298_ enthalpies of the most stable conformers were converted into the gas-phase standard molar enthalpies of formation,$\;\Delta_{f}H_{m}^{o}$(g), using the *experimental* gas phase standard molar enthalpies of formation $\Delta_{f}H_{m}^{o}$(g, 298.15 K) of auxiliary compounds (see Appendix A) using the following well-balanced reactions (WBR):*x-hydroxy-benzophenone* + *benzene* = *hydroxybenzene* + *benzophenone*(9)
*x-methoxy-benzophenone* + *benzene* = *methoxybenzene* + *benzophenone*(10)
*x,x-dihydroxy-benzophenone* + *2* × *benzene* = *2* × *hydroxybenzene* + *benzophenone*(11)
*x,x-dihydroxy-x-methoxybenzophenone* + 3 × *benzene* =*2* × *hydroxybenzene* + *methoxybenzene* + *benzophenone*(12)
*x,x,x,x-tetrahydroxy-benzophenone* + 4 × *benzene* = 4 × *hydroxybenzene* + *benzophenone*(13)
*x,x-dihydroxy-x,x-dimethoxybenzophenone* + *4* × *benzene* =*2* × *hydroxybenzene* + *2* × *methoxybenzene* + *2* × *benzophenone*(14)

The results calculated with the WBR,$\;\Delta_{f}H_{m}^{o}$(g)_WBR_, are given in Table 8, column 5. In addition, the *H*_298_ enthalpies of the most stable conformers were converted into the enthalpies of formation with the atomisation reaction (AT) and these results,$\;\Delta_{f}H_{m}^{o}$(g)_AT_, are given in Table 8, column 6. As can be seen from Table 8, the results for $\Delta_{f}H_{m}^{o}$(g)_WBR_ and $\Delta_{f}H_{m}^{o}$(g)_AT_ are practically indistinguishable. Therefore, we averaged these results to develop the *theoretical* enthalpies of formation, $\Delta_{f}H_{m}^{o}$(g)_theor_, for each molecule (see Table 8, column 7).

The *theoretical* enthalpies of formation of substituted benzophenones calculated using the G4 method agree only sufficiently with the experiment (within the boundaries of their combined uncertainties). However, it should be mentioned that the combustion experiments had some deficiency as follows. The three samples of hydroxy-benzophenones studied by Davalos et al. [15] were only 99.2% pure according to DSC. A possible residual amount of water traces in samples was not characterised either. Moreover, the commercial sample of 3-hydroxy-benzopenone was measured without additional purification. The commercial sample of 2-hydroxy-4-methoxy-benzopenone was measured by Lago et al. [25] also without additional purification and attestation of traces of water. Finally, no details on the sample purity of 2,4-di-hydroxybenzophenone and on the experimental conditions can be found in the publication by Contineanu and Marchidan [41], therefore the significant deviation from the quantum chemical result shown in Table 8 should be regarded as an indicator for the need for additional combustion experiments with this compound. Considering the shortcomings of the experimental $\Delta_{f}H_{m}^{o}$(g)-data compiled in Table 8, we decided to use the consistent set of *theoretical* enthalpies of formation for substituted benzophenones for the development of a “centrepiece” group-contribution approach as follows. 

## 4. Development and Practical Application of the “Centrepiece” Group-Contribution Approach

### 4.1. Construction of a Theoretical Framework

The “centrepiece” approach has been described in detail in our previous papers [27,36,45]. The basic idea of the “centrepiece approach” approach is to select a relatively large “centrepiece molecule” (rather than the traditional summation of group contributions) with well-known thermodynamic properties that structurally most closely resembles the molecule of interest. Related to the compounds discussed in this paper, the benzophenone itself is the most suitable “centrepiece” molecule. Various substituents (hydroxyl and methoxy in this work) can be attached to the “centrepiece” at different positions on the benzene rings of the benzophenone (see Figure 4).

The enthalpic contributions for these substituents can be easily derived (see Figure 5) from the differences between the enthalpy of the substituted benzene and the enthalpy of the benzene itself. 

Using this scheme, the required for this work contributions Δ*H*(H *→ OH*) and Δ*H*(H *→ CH_3_O*) were derived (see Table 9) using the reliable thermochemical data for benzene, methoxybenzene, and hydroxybenzene compiled in Appendix A. We postulate that these contributions are applicable to any aromatic ring system. Hence, the enthalpic contributions Δ*H*(H *→ OH*) and Δ*H*(H *→ CH_3_O*) can now be applied to construct a framework of any hydroxy- and methoxy-substituted benzophenone, starting with the benzophenone as the “centrepiece” (see Figure 4).

As a rule, this framework can energetically predict at a rough level the vaporisation or formation enthalpies. However, this framework is not perfect since it lacks the energetics of the interactions between the carbonyl and the substituents attached to the phenyl rings of benzophenone. For a more accurate assessment, the pairwise nearest and non-nearest neighbour interactions of substituents on the “centrepiece” framework should also be considered as follows.

### 4.2. Pairwise Interactions of Substituents on the Benzene Ring

The non-nearest neighbour (e.g., *meta*- or *para*- interactions) or nearest neighbour (e.g., *ortho*-interactions) interactions of substituents on the benzene ring are an indispensable part of the energetics of aromatic molecules. However, quantitatively they are dependent on the type and position of the substituent. As a rule, *meta*- or *para*- pairwise interactions are weak, and *ortho*- interactions are powerful. How the pairwise interactions were derived is shown in Figure 6. 

For example, to quantify the enthalpic contribution “*meta C = O(C_6_H_5_) − OH*” responsible for the non-bonded interaction of the carbonyl and *OH*-group in the *meta*-position on benzophenone (taken as the “centrepiece”), we must first construct the “theoretical framework” of 3-hydroxy-benzophenone (see Figure 6). To do that, we simply add the contribution Δ*H*(H *→ OH*) from Table 9 to the experimental enthalpy (enthalpy of vaporisation or enthalpy of formation) of the benzophenone (as a “centrepiece”) also given in Table 9. This “theoretical framework” of 3-hydroxy-benzophenone does not contain the “*meta C = O(C_6_H_5_) − OH*” interaction. However, this interaction is present in the real 3-hydroxy-benzophenone (it is symbolised in Figure 6 with a red arrow). The arithmetic difference between the experimental enthalpy of 3-hydroxy-benzophenone and the enthalpy of the “theoretical framework” therefore provides the quantitative size of the pairwise interaction “*meta C = O(C_6_H_5_) − OH*” directly (see Table 9). Using the same logic, the enthalpic contributions for the “*ortho C = O(C_6_H_5_) − OH*” and “*para C = O(C_6_H_5_) − OH*” were derived from data for 2-hydroxy- and 4-hydroxy-benzophenone. In the same way, the required enthalpy contributions for pairwise interactions of carbonyl and methoxy substituent were estimated and summarised in Table 9.

### 4.3. Practical Application of the Centerpiece Approach for Prediction of Enthalpies of Substituted Benzophenones

As can be seen from Table 9, the magnitudes of the pairwise interactions in terms of $\Delta_{l}^{g}H_{m}^{o}$ are mostly not negligible. However, meaningful discussion of these magnitudes is rather limited since these contributions reflect the tightness of molecular packing in the liquid. They must be considered as empirical constants for the correct prediction of the vaporisation energetics. Does it work? We demonstrate the principle applicability of the “centrepiece” approach in the case that at least two substituents are attached to the benzophenone as a “centrepiece”. In Figure 7, the algorithm for calculating the vaporisation enthalpy of the 2,4-dihydroxy-benzophenone using the “centrepiece” approach with the numerical values from Table 9 is shown.

It was found that the empirical enthalpy of vaporisation of 2,4-dihydroxy-benzophenone, $\Delta_{l}^{g}H_{m}^{o}$ (298.15 K)_theor_ = 113.0 kJ·mol^−1^, agrees well with the experimental value,$\;\Delta_{l}^{g}H_{m}^{o}$ (298.15 K) = 112.3 ± 4.2 kJ·mol^−1^, evaluated in Table 4. Therefore, the “centrepiece” approach can be successfully applied to predict the vaporisation enthalpies of substituted benzophenones and other aromatic systems with the contributions derived in Table 9. We now apply the “centrepiece” approach to predicting the gas-phase enthalpies of formation of substituted benzophenones.

It makes more sense to discuss the magnitudes of the pairwise interactions with respect to $\Delta_{f}H_{m}^{o}$(g) given in Table 9, since these non-covalent interactions are generally responsible for the distribution of electron density in the molecule. Moreover, they can be used to derive the strength of the intra-molecular hydrogen bonding present in the 2-hydroxy-substituted benzophenones (see Section 5).

Quantitatively, the intensity of the non-covalent interactions depends strongly on the nature of the *ortho*-, *meta*-, or *para*-pairs. It can be seen from Table 9 that in terms of $\Delta_{f}H_{m}^{o}$(g) the *ortho*-hydroxy-benzophenone shows a strong stabilization of −24.5 kJ·mol^−1^ due to intra-molecular hydrogen bonding. In contrast, the *ortho*-dihydroxy-benzene shows a destabilization of 2.5 kJ·mol^−1^ despite the stabilizing intra-molecular hydrogen bonding present in this molecule. The *ortho*-dimethoxy-benzene also shows a destabilization of 17.5 kJ·mol^−1^ due to the sterical repulsion of bulky methoxy groups. In our experience, the *meta*- and *para*-interactions of substituents on the benzene ring are less profound compared to *ortho*-interactions [39,40]. Indeed, the *meta*- and *para*-interactions of the OH and CH_3_O substituents with the *carbonyl* group can be considered as weak, being below 4 kJ·mol^−1^ (see Table 9). In contrast, the strong destabilization of 11.4 kJ·mol^−1^ is observed for the *para*-isomer of methoxy-phenol. Moreover, a significant destabilization of 7.3 kJ·mol^−1^ is observed for the *para*-isomer of dimethoxy-benzene. These noticeable destabilizing effects can be explained by the specific electron density distribution within the substituted benzene ring. 

Since some of the pairwise non-covalent interactions in terms of $\Delta_{f}H_{m}^{o}$(g) have been derived using substituted benzenes (see Table 9), their applicability to benzophenone derivatives needs to be checked. We calculated the enthalpy of formation of the 2-hydroxy-4-methoxy-benzophenone using the “centrepiece” approach with the numerical values from Table 9. The calculation algorithm is given in Figure 8.

It was found that the *empirical* enthalpy of formation of 2-hydroxy-4-methoxy-benzophenone, $\Delta_{f}H_{m}^{o}$ (g, 298.15 K) = −306.2 kJ·mol^−1^, agrees with the experimental value, $\Delta_{f}H_{m}^{o}$ (g, 298.15 K) = −297.4 ± 4.7 kJ·mol^−1^ [25]. Therefore, the “centrepiece” approach can also be used successfully to predict the enthalpies of formation of substituted benzophenones and other aromatic systems with the contributions derived in Table 9.

## 5. Strength of Intra-Molecular Hydrogen Bonding in Ortho-Substituted Benzophenones

In this section, we consider the intra-molecular hydrogen bonded 2-hydroxy substituted benzophenones. It is known from XRD data that the distance between the carbonyl oxygen O(1) and the hydrogen of OH group (see Figure 9, left) in 2-hydroxy-benzophenone is 1.810 Å in the crystal state [15,47], and 1.678 Å in the gas state, as calculated at the B3LYP/6-311++G(3df,2p) level [15], and 1.75 Å as calculated with G3MP2 in this work.

In 2,2’-di-hydroxy-benzophenone both hydroxyl groups act as intramolecular hydrogen-bond donors to the O(1) carbonyl atom (see Figure 9, right). From XRD data in the crystal state, the distance between the carbonyl oxygen O(1) and the hydrogen of H-O(2) group is 1.873 Å and between the carbonyl oxygen O(1) and the hydrogen of H-O(3) group is 1.772 Å [48]. The reason for the difference in the two intra-molecular hydrogen bonds in 2,2’-di-hydroxy-benzophenone is not evident. In the shorter one, O(1)…H-O(3), the hydroxyl group is also involved in a bifurcated *inter*-molecular interaction [48]. The other hydroxyl group H-O(2) therefore appears to be somewhat deficient in hydrogen bonding [48]. In the gas phase, however, the distances between the carbonyl oxygen O(1) and both hydroxyl groups become indistinguishable: 1.729 Å and 1.728 Å as calculated at the B3LYP-6-311++G** level [49]. In this work, the distances of 1.79 Å and 1.79 Å were calculated with G3MP2.

In a qualitative way, intra-HB strength can be assessed in terms of OH chemical shifts [50], double-bond deuterium isotope effects on ^13^C chemical shifts [51], OH stretch frequencies [52], or O…O distances [53]. The quantitative way to determine intra-HB strength is a challenging task because, as shown below, the choice of an appropriate “non-bonded” reference system is thwarted with complications.

### 5.1. Strength of Intra-Molecular Hydrogen Bonding from the “Ortho-Para” Method

Determination of the intra-molecular hydrogen bonding strength in 2-hydroxy-benzophenone using the “ortho-para” method is illustrated in Figure 10. 

This approach was suggested by Minas da Piedade [54]. The idea behind it is to use a simple *“ortho-para”* isomerization reaction, provided that experimental $\Delta_{f}H_{m}^{o}$(g)-values for the reaction participants are known. It is obvious that the intra-HB is present on the left side of the reaction and absent on the right side. Therefore, the enthalpy of this reaction should represent the intra-HB strength. Using the experimental $\Delta_{f}H_{m}^{o}$(g)-values for 2-hydroxy-benzophenone and 4-hydroxy-benzophenone given in Table 8, the intra-HB strength was calculated to be (−147.8 − 122.1) = −25.7 ± 5.7 kJ·mol^−1^. However, this result should be corrected with the additional interaction between the OH and carbonyl group, as *para C = O*(*C_6_H_5_*) − *OH* = −3.4 kJ·mol^−1^ (see Table 9, column 2), present in 4-hydroxy-benzophenone. Only now the result (−25.7 + 3.4) = −22.3 ± 5.7 kJ·mol^−1^ could be considered as the strength of the intra-HB in 2-hydroxy-benzophenone. Unfortunately, this “corrected” *ortho-para* method can only be used to determine intra-HB strength in 2-hydroxybenzophenone. The structures of other ortho-substituted benzophenones (see Figure 2) are too complex to find a suitable reference molecule.

### 5.2. Strength of Intra-Molecular Hydrogen Bonding from the Well-Balanced Reactions

In our recent work, we developed thermodynamically based tools to quantify HB strength in aliphatic [50] and aromatic systems [55]. One of them is based on the enthalpies of *well-balanced reactions* (9 to 14) as it was already shown in Section 4. The idea behind this is shown in Figure 11, where reaction (9) is written in reverse.

The enthalpies of well-balanced reactions (9 to 14) are given in Table 10, column 2.

In contrast to the “*ortho-para*” method, the reference molecules for this method do not contain any additional substituent interactions that would have to be corrected. Indeed, in the case of 2-hydroxy-benzophenone shown in Figure 11, the enthalpy of this reaction is equivalent to the pairwise interaction of carbonyl and hydroxyl groups, defined as *ortho C = O*(*C_6_H_5_*) − *OH* = −24.5 kJ·mol^−1^ in Section 4 (see Table 9, column 2). The latter interaction could be considered as the strength of the intra-HB in 2-hydroxy-benzopenone and agrees well with those −22.3 ± 5.7 kJ·mol^−1^, derived from the “*ortho-para*” method.

Also, for 2,2’-di-hydroxy-benzophenone, the enthalpy of the well-balanced reaction is directly related to the intra-HB-strength Δ*H*_WBR_ = −20.4 kJ·mol^−1^ per single bond (see Figure 12).

The intra-HB strength in 2,2’-dihydroxybenzophenone is slightly less than in 2-hydroxy-benzophenone, although the distances between the O(1) and hydroxy groups are shorter in 2,2’-di-hydroxy-benzophenone (as calculated using G3MP2 in this work).

For other di-hydroxy- and tetra-hydroxy-substituted benzophenones given in Table 10, the enthalpies of the well-balanced reactions require some correction before being related to the intra-HB strength. For example, as shown in Figure 13, in 2,4-dihydroxybenzophenone we need to subtract the *para C = O*(*C_6_H_5_*) − *OH* and *meta OH*–*OH* interactions given in Table 9. The resulting intra-HB-strength Δ*H*_WBR_ = −24.3 kJ·mol^−1^ (see Table 10, column 3) can hardly be distinguished from those in 2-hydroxybenzophenone. Similarly, we “corrected” the well-balanced reaction enthalpies (see Appendix A) and the resulting strengths are shown in Table 10, column 3.

It has turned out that the intra-HB strength in mono-, di-, tri-, and tetra- substituted benzophenones is generally at a similar level of −24 kJ·mol^−1^ and the influence of the nearest neighbour substituents is about 5 kJ·mol^−1^ irregular decrease or increase in intra-HB strength.

### 5.3. Strength of Intra-Molecular Hydrogen Bonding from the “HB and Out” Method

With the modern development of quantum chemical (QC) methods, the quantitative “theoretical” measure of intra-HB strength is commonly defined as the energy difference between the hydrogen-bonded and open conformer (OH group rotated 180 °C around the C–O axis). This method is referred to as “HB and Out” [56] but requires that the OH group of the open conformer is not involved in steric or other interfering interactions [57].

2-Hydroxy-substituted benzophenones are very suitable molecules for studying of hydrogen bond energies estimated by the “HB and Out” method as the OH proton has no noticeable interactions in the open “Out” conformations. 

The magnitudes of the intra-HB strength of 2-hydroxy- and 2,2’-dihydroxy-substituted benzophenones were estimated by subtracting the QC computed energies of the non-hydrogen-bonded conformer from the energies of the hydrogen-bonded conformer, shown in Figure 3. These values are referred to as Δ*H*_conf_ and are summarised in Table 10 for comparison. For 2-hydroxy-benzophenone Δ*H*_conf_ values were calculated using both the G4 and the G3MP2 methods (see Figure 3). It has turned out that the results are practically indistinguishable. Taking into account that the G3MP2 method is significantly less time-consuming compared to the G4 method, all Δ*H*_conf_-values in Table 10 were calculated using the G3MP2.

The strength of intra-HB in 2-hydroxy-benzophenone calculated by the G3MP2 method is −35.7 kJ·mol^−1^. The strength of two intra-HBs in 2,2’-di-hydroxy-benzophenone calculated by the G3MP2 method is −68.6 kJ·mol^−1^, but the strength related to a single HB is −34.3 kJ·mol^−1^ and is equal to that in 2-hydroxy-benzophenone. These results differ significantly from the intra-HB strength in 2-hydroxy-benzophenone, −24.5 kJ·mol^−1^, and in 2,2’di-hydroxy-benzophenone, −40.8 kJ·mol^−1^, as derived from the “*ortho-para*” method and from WBR. Moreover, for other substituted benzophenones the strength estimated with “HB and Out” is systematically more negative with the differences shown in Table 10, last column. 

Which values are correct? Apparently, it is the coinciding values derived with the “*ortho-para*” and WBR methods! This is because the consistency of the experimental and quantum chemical results for substituted benzophenones has been convincingly demonstrated in the previous sections. Moreover, the hydrogen bonded conformations of the ortho-hydroxy-substituted benzophenones were involved in calculations by the WBR method, confirming the level of the intra-HB strength evaluated by these methods. From these facts, it can be concluded that the reason for the observed discrepancy could therefore rather lie in the definition of the intra-HB strength by the “HB and Out” method.

Does the intra-HB strength, defined as the energy difference between the hydrogen-bonded and open forms (OH group rotated 180° around the C–O axis), really represent a correct “non-H-bonded” reference? Perhaps an alternative to the “HB and Out” procedure, the phenyl group can be rotated 180° about the C–C axis (see Figure 14).

We calculated the energy of the later conformation using G3MP2 and the destabilisation of 38.1 kJ·mol^−1^ is clear evidence that in both non-H-bonded “reference” conformations there is a significant amount of sterical repulsions of the OH-group and the phenyl ring is incorporated, and these conformations are hardly suitable to be used as references for determining the strength of the intra-HB strength in hydroxy-substituted benzophenones. Consequently, only the “ortho-para” and WBR methods should be applied for quantification of intra-HB strength in similarly shaped molecules.

## 6. Conclusions

The consistent sets of standard molar thermodynamic properties of formation and phase transitions for substituted benzophenones were evaluated in this work with help of complementary measurements of vapour pressures, sublimation/vaporisation, and fusion enthalpies, as well as with help of empirical and high-level quantum chemical calculations. Thermodynamic properties of substituted benzophenones were recommended as reliable benchmark properties for thermochemical calculations. The evaluated vaporisation and formation enthalpies were used to design and develop the “centrepiece” approach for prediction of thermodynamic properties of the aromatic systems. The strength of intra-molecular hydrogen bonding was evaluated using quantum chemical and thermochemical methods.

## Figures and Tables

**Figure 1 molecules-27-08477-f001:**
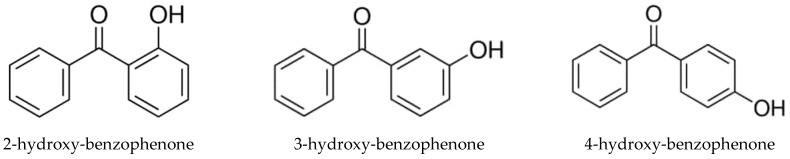
Mono-substituted benzophenones studied in this work.

**Figure 2 molecules-27-08477-f002:**
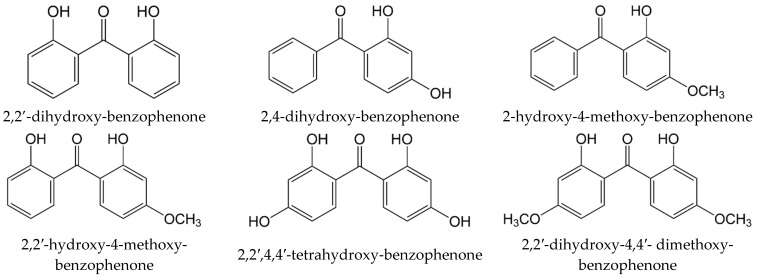
Poly-substituted benzophenones studied in this work.

**Figure 3 molecules-27-08477-f003:**
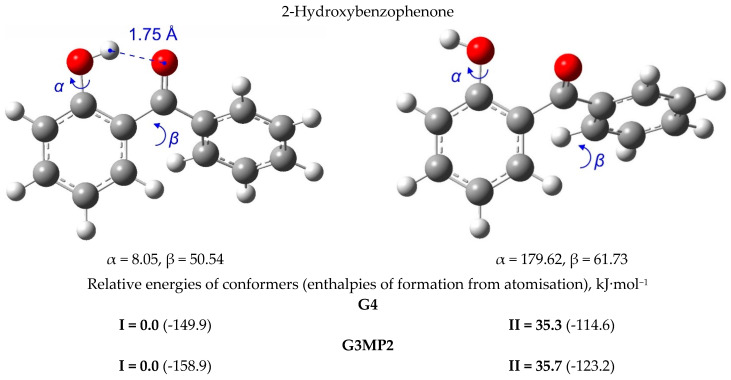
The stable conformers of 2-hydroxy-substituted benzophenones.

**Figure 4 molecules-27-08477-f004:**
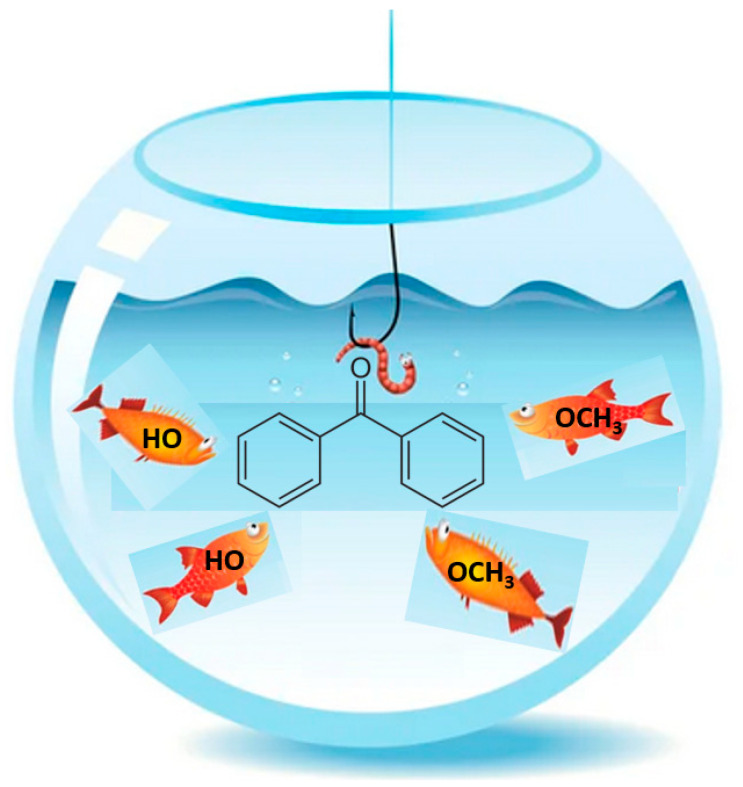
Graphical presentation of the idea of a “centrepiece” group-contribution approach.

**Figure 5 molecules-27-08477-f005:**
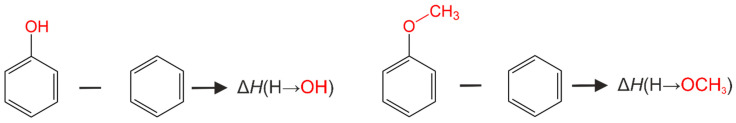
Quantification of the enthalpic contributions for the hydroxy- and methoxy substituents. The scheme is valid for calculations with the standard molar enthalpies of vaporisation, as well as for the calculations with the gas-phase standard molar enthalpies of formation.

**Figure 6 molecules-27-08477-f006:**
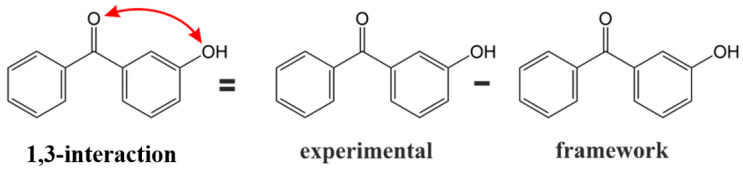
Example for a quantification of the 1,3-non-nearest neighbour interactions of the carbonyl-group with the OH substituent in 3-hydroxy-benzophenone. The scheme applies both to the calculation of the standard molar enthalpies of vaporisation, and to the calculation of the standard molar enthalpies of formation in the gas-phase.

**Figure 7 molecules-27-08477-f007:**
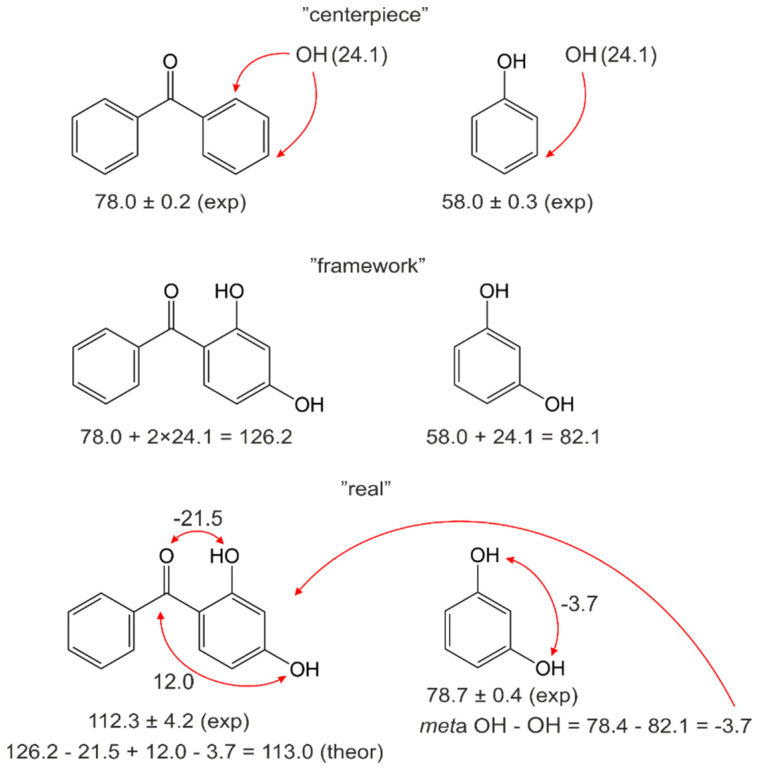
Example of calculation of the enthalpy of vaporisation, $\Delta_{l}^{g}H_{m}^{o}$ (298.15 K), of 2,4-dihydroxy-benzophenone using the “centerpiece” approach with the numerical values from Table 9. All numbers are given in kJ·mol^−1^.

**Figure 8 molecules-27-08477-f008:**
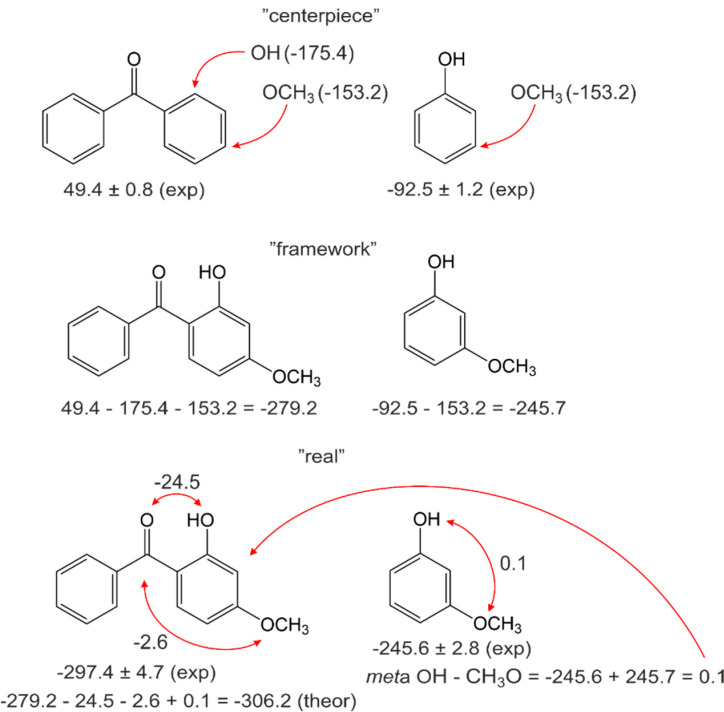
Example of calculation of the enthalpy of formation, $\Delta_{f}H_{m}^{o}$(g, 298.15 K), of 2-hydroxy-4-methoxy-benzophenone using the “centrepiece” approach with the numerical values from Table 9. All numbers are given in kJ·mol^−−1^.

**Figure 9 molecules-27-08477-f009:**
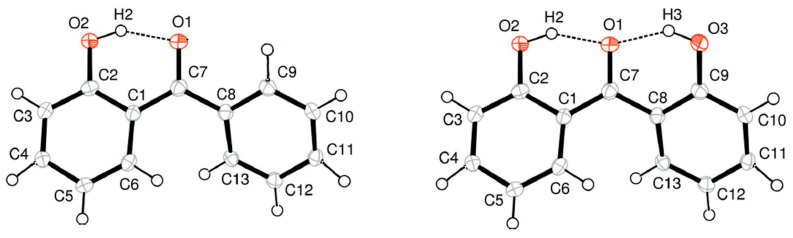
Intra-molecular hydrogen bonding in 2-hydroxy-benzophenone (**left**) and 2,2’-di-hydroxy-benzophenone (**right**).

**Figure 10 molecules-27-08477-f010:**
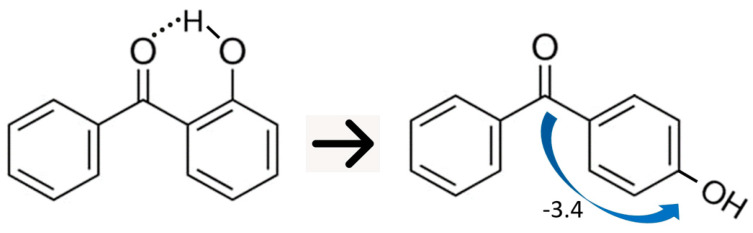
Determination of the intra-molecular hydrogen bonding strength in 2-hydroxy-benzophenone using the “ortho-para” method. The blue arrow represents the additional *para C = O*(*C_6_H_5_*) − *OH* = −3.4 kJ·mol^−1^, that should be subtracted from the enthalpy of this reaction to attribute the corrected result to intra-HB strength.

**Figure 11 molecules-27-08477-f011:**
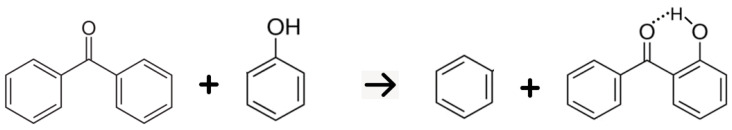
Determination of the intra-molecular hydrogen bonding strength in 2-hydroxy-benzophenone using the “*well-balanced reaction*” method. The reaction enthalpy $\Delta_{r}H_{m}^{o}$_(WBR)_ = −24.5 kJ·mol^−1^ (see Table 10, column 2). The intra-HB-strength Δ*H*_WBR_ = −24.5 kJ·mol^−1^ (see Table 10, column 3).

**Figure 12 molecules-27-08477-f012:**
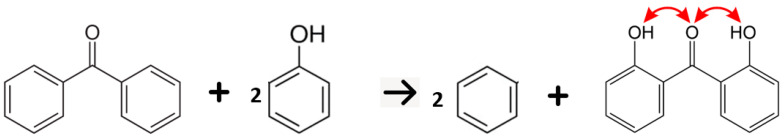
Determination of the intra-molecular hydrogen bonding strength in 2-hydroxy-benzophenone using the “*well-balanced reaction*” method. The reaction enthalpy $\Delta_{r}H_{m}^{o}$_(WBR)_ = −40.8 kJ·mol^−1^ (see Table 10, column 2). The intra-HB-strength Δ*H*_WBR_ = −40.8/2 = −20.4 kJ·mol^−1^ per single bond.

**Figure 13 molecules-27-08477-f013:**
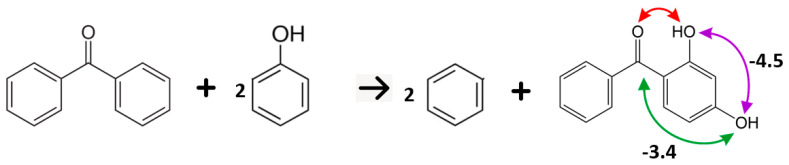
Determination of the intra-molecular hydrogen bonding strength in 2,4-di-hydroxy-benzophenone using the “well-balanced reaction” method. The reaction enthalpy $\Delta_{r}H_{m}^{o}$_(WBR)_ = −32.2 kJ·mol^−1^ (see Table 10, column 2). Additional interactions *para C = O*(*C_6_H_5_*) − *OH* = −3.4 kJ·mol^−1^ and *meta OH*–*OH* = −4.5 kJ·mol^−1^ (see Table 9). The intra-HB-strength Δ*H*_WBR_ = (−32.2 + 3.4 + 4.5) = −24.3 kJ·mol^−1^ (see Table 10, column 3).

**Figure 14 molecules-27-08477-f014:**
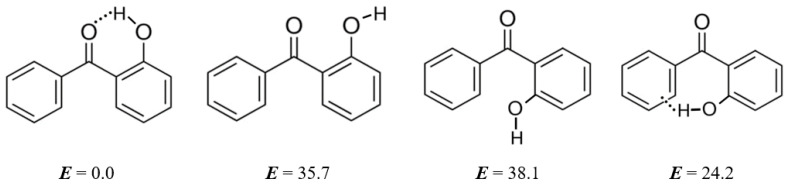
Determination of the intra-molecular hydrogen bonding strength in 2-hydroxy-benzophenone using the “HB and Out” method and an alternative phenyl group rotated 180° around the C-C axis. (All values in kJ·mol^−1^).

**Table 1 molecules-27-08477-t001:** Results of transpiration method for substituted benzophenones: absolute vapour pressures *p*, standard molar vaporisation/sublimation enthalpies, and standard molar vaporisation/sublimation entropies.

*T*/ K ^a^	*m*/ mg ^b^	*V*(N_2_) ^c^/dm^3^	*T*_a_/ K ^d^	Flow/ dm^3^·h^−1^	*p*/ Pa ^e^	*u*(*p*)/ Pa ^f^	$\Delta_{l,cr}^{g}H_{m}^{o}(T)/$ kJ·mol^−1^	$\Delta_{l,cr}^{g}S_{m}^{o}(T)/$ J·K^−1^·mol^−1^
2,2′-di-hydroxy-benzophenone$:\;\Delta_{l}^{g}H_{m}^{o}$ (298.15 K) = (84.9 ± 0.9) kJ·mol^−1^ $\ln\;\left(p/p_{ref}\right)=\frac{375.0}{R}-\frac{121793.3}{RT}-\frac{123.6}{R}\ln\frac{T}{298.15}$; *p*_ref_ = 1 Pa								
363.1	0.69	1.274	295.7	2.94	6.18	0.18	76.9	131.3
368.2	1.55	2.065	295.9	1.18	8.62	0.24	76.3	129.4
363.2	1.31	2.439	295.2	1.18	6.14	0.18	76.9	131.1
368.6	1.29	1.666	295.9	1.02	8.89	0.25	76.2	129.3
373.5	1.62	1.475	295.9	1.18	12.59	0.34	75.6	127.8
373.6	0.53	0.479	295.7	1.07	12.72	0.34	75.6	127.8
375.8	1.31	1.014	295.9	1.01	14.85	0.40	75.3	127.2
378.7	1.71	1.088	295.9	2.04	18.05	0.48	75.0	126.3
378.8	1.52	0.998	295.9	1.02	17.51	0.46	75.0	126.0
383.8	1.53	0.715	295.9	1.02	24.56	0.64	74.4	124.7
383.8	1.55	0.710	295.2	1.01	24.99	0.65	74.4	124.8
383.8	1.29	0.610	295.2	1.18	24.30	0.63	74.4	124.5
387.9	1.25	0.457	295.2	1.14	31.23	0.81	73.8	123.3
388.1	1.09	0.408	295.9	0.98	30.58	0.79	73.8	123.0
388.7	1.80	0.633	295.2	1.00	32.53	0.84	73.8	123.0
393.0	1.02	0.286	295.2	1.14	40.81	1.05	73.2	121.4
393.1	1.33	0.372	295.2	1.01	41.09	1.05	73.2	121.4
398.1	1.36	0.291	295.7	1.03	53.59	1.36	72.6	119.7
398.3	1.15	0.237	295.9	1.02	55.80	1.42	72.6	119.9
403.2	1.72	0.274	295.7	1.03	71.82	1.82	72.0	118.3
408.4	2.11	0.257	295.2	1.03	93.81	2.37	71.3	116.7
2-hydroxy-4-methoxy-benzophenone$:\;\Delta_{l}^{g}H_{m}^{o}$ (298.15 K) = (98.4 ± 1.0) kJ·mol^−1^ $\ln\;\left(p/p_{ref}\right)=\frac{396.8}{R}-\frac{134317.1}{RT}-\frac{120.5}{R}\ln\frac{T}{298.15}$; *p*_ref_ = 1 Pa								
341.2	0.16	8.480	299.3	5.09	0.207	0.010	93.2	164.4
343.2	0.15	6.716	296.3	5.04	0.243	0.011	93.0	163.4
346.2	0.15	5.014	300.4	5.01	0.335	0.013	92.6	162.7
348.2	0.05	1.329	298.6	2.05	0.397	0.015	92.4	161.9
348.2	0.15	4.070	299.4	5.09	0.404	0.015	92.4	162.0
350.2	0.18	4.057	298.6	3.01	0.476	0.017	92.1	161.2
350.2	0.15	3.562	299.4	5.09	0.474	0.017	92.1	161.2
353.2	0.15	2.667	295.4	3.02	0.608	0.020	91.8	160.0
353.2	0.15	2.714	299.3	5.09	0.620	0.020	91.8	160.1
357.2	0.16	2.015	298.8	5.04	0.867	0.027	91.3	158.7
358.2	0.10	1.107	295.4	3.02	0.958	0.029	91.2	158.5
359.2	0.14	1.503	298.6	3.01	1.025	0.031	91.0	158.0
359.2	0.15	1.611	301.3	5.09	1.027	0.031	91.0	158.0
360.2	0.15	1.511	299.4	5.04	1.108	0.033	90.9	157.6
362.1	0.15	1.208	299.8	3.02	1.337	0.038	90.7	157.2
363.2	0.16	1.180	301.8	3.08	1.450	0.041	90.6	156.7
365.1	0.15	0.906	300.3	3.02	1.760	0.049	90.3	156.4
366.2	0.16	0.923	301.8	3.08	1.887	0.052	90.2	155.9
367.2	0.18	0.981	300.4	3.02	2.042	0.056	90.1	155.5
368.2	0.14	0.719	299.4	2.98	2.190	0.060	90.0	155.1
2,2′-di-hydroxy-4-methoxy-benzophenone$:\;\Delta_{l}^{g}H_{m}^{o}$ (298.15 K) = (100.7 ± 0.9) kJ·mol^−1^ $\ln\;\left(p/p_{ref}\right)=\frac{412.8}{R}-\frac{141902.1}{RT}-\frac{138.3}{R}\ln\frac{T}{298.15}$; *p*_ref_ = 1 Pa								
365.1	0.36	5.608	294.7	3.06	0.64	0.02	91.4	151.0
370.2	0.35	3.824	294.2	3.06	0.93	0.03	90.7	148.7
375.0	0.38	2.679	293.7	2.92	1.44	0.04	90.0	147.4
375.3	2.15	14.89	294.2	1.02	1.45	0.04	90.0	147.2
380.3	0.37	1.861	295.2	3.06	1.98	0.05	89.3	144.9
385.3	1.51	5.243	293.2	3.15	2.87	0.08	88.6	143.1
390.3	0.42	1.011	295.9	1.17	4.18	0.11	87.9	141.5
390.4	1.40	3.220	293.7	3.12	4.36	0.11	87.9	141.7
390.7	0.48	1.122	295.2	1.87	4.31	0.11	87.9	141.4
395.4	1.13	1.940	294.7	3.15	5.85	0.17	87.2	139.6
400.2	1.30	1.558	293.7	3.12	8.35	0.23	86.6	138.2
400.4	0.38	0.466	295.9	1.17	8.10	0.23	86.5	137.8
400.7	0.43	0.504	294.7	0.98	8.57	0.24	86.5	138.0
405.4	1.30	1.170	294.7	3.05	11.17	0.30	85.8	136.0
405.4	1.30	1.170	294.7	3.05	11.17	0.30	85.8	136.0
410.6	1.61	0.987	293.7	3.12	16.32	0.43	85.1	134.8
410.9	1.31	0.818	293.7	2.34	16.07	0.43	85.1	134.4
415.5	1.49	0.691	294.2	1.93	21.61	0.57	84.4	133.1
415.9	1.45	0.651	294.7	1.95	22.39	0.58	84.4	133.1
415.9	1.43	0.647	293.7	1.02	22.17	0.58	84.4	133.0
420.9	1.38	0.476	294.2	1.02	28.95	0.75	83.7	131.1
420.9	1.38	0.488	295.2	1.95	28.48	0.74	83.7	130.9
425.9	1.20	0.315	294.2	0.99	38.20	0.98	83.0	129.5
430.9	1.15	0.232	295.2	0.99	49.83	1.27	82.3	127.8
2,2′-di-hydroxy-4,4′-di-methoxy-benzophenone$:\;\Delta_{l}^{g}H_{m}^{o}$ (298.15 K) = (117.2 ± 2.9) kJ·mol^−1^ $\ln\;\left(p/p_{ref}\right)=\frac{452.8}{R}-\frac{162824.5}{RT}-\frac{153.0}{R}\ln\frac{T}{298.15}$; *p*_ref_ = 1 Pa								
412.6	0.58	1.914	296.2	3.06	2.74	0.07	99.7	154.3
415.0	1.11	3.062	294.2	3.06	3.24	0.09	99.3	153.4
420.0	1.98	3.776	295.2	3.06	4.69	0.12	98.6	151.8
424.0	1.74	2.552	295.2	3.06	6.10	0.18	97.9	150.3
435.1	1.68	1.250	293.7	3.06	11.96	0.32	96.3	146.2
2,2′-di-hydroxy-4,4′-di-methoxy-benzophenone$:\;\Delta_{\mathrm{cr}}^{g}H_{m}^{o}$ (298.15 K) = (139.4 ± 2.4) kJ·mol^−1^ $\ln\;\left(p/p_{ref}\right)=\frac{401.7}{R}-\frac{155110.8}{RT}-\frac{52.8}{R}\ln\frac{T}{298.15}$; *p*_ref_ = 1 Pa								
368.5	0.28	96.12	294.2	6.07	0.0258	0.0056	135.7	242.0
372.9	0.54	108.5	295.9	6.00	0.0443	0.0061	135.4	241.5
375.0	0.56	87.94	295.9	6.00	0.0575	0.0064	135.3	241.3
376.0	0.30	42.31	295.9	2.82	0.0628	0.0066	135.3	241.1
376.7	0.31	43.12	295.9	3.06	0.0650	0.0066	135.2	240.5
378.9	0.24	25.30	294.2	6.07	0.0849	0.0071	135.1	240.4
382.9	0.27	18.00	295.9	6.00	0.1361	0.0084	134.9	240.0
383.0	0.27	18.00	292.7	6.00	0.1312	0.0083	134.9	239.6
386.4	0.31	14.42	293.7	6.07	0.1916	0.0098	134.7	239.2
389.0	0.22	7.504	293.7	6.00	0.2663	0.0117	134.6	239.3
392.1	0.34	8.704	295.9	6.00	0.3509	0.0138	134.4	238.4
395.1	0.35	6.100	295.9	6.00	0.5143	0.0179	134.3	238.6
399.3	0.28	3.137	293.7	6.07	0.7853	0.0246	134.0	237.9

^a^ Saturation temperature measured with the standard uncertainty (*u*(*T*) = 0.1 K). ^b^ Mass of transferred sample condensed at *T* = 243 K. ^c^ Volume of nitrogen (*u*(*V*) = 0.005 dm^3^) used to transfer *m* (*u*(*m*) = 0.0001 g) of the sample. Uncertainties are given as standard uncertainties. ^d^
*T*_a_ is the temperature of the soap bubble meter used for measurement of the gas flow. ^e^ Vapour pressure at temperature *T*, calculated from the *m* and the residual vapour pressure at the condensation temperature calculated by an iteration procedure. ^f^ Uncertainties were calculated with *u*(*p_i_*/Pa) = 0.005 + 0.025 (*p_i_*/Pa) for pressures below 5 Pa and with *u*(*p_i_*/Pa) = 0.025 + 0.025(*p_i_*/Pa) for pressures from 5 to 3000 Pa. The standard uncertainties for *T*, *V*, *p*, *m*, are standard uncertainties with 0.683 confidence level. Uncertainties of the sublimation/vaporisation enthalpies are expressed as the expanded uncertainty (0.95 level of confidence, k = 2). They were calculated according to a procedure described elsewhere [13,14]. They include uncertainties from transpiration experimental conditions, uncertainties of vapour pressure, uncertainties from the fitting equation, and uncertainties from temperature adjustment to *T* = 298.15 K.

**Table 2 molecules-27-08477-t002:** Results of Knudsen effusion method for benzophenone derivatives: absolute vapour pressures *p*, standard molar sublimation enthalpies and standard molar sublimation entropies.

*T*/ K ^a^	*m*/ mg ^b^	*T/*s	*p*/ Pa ^c^	*u*(*p*)/ Pa ^d^	$\Delta_{cr}^{g}H_{m}^{o}(T)/$ kJ·mol^−1^	$\Delta_{cr}^{g}S_{m}^{o}(T)/$ J·K^−1^·mol^−1^
2,4-di-hydroxy-benzophenone$:\;\Delta_{\mathrm{cr}}^{g}H_{m}^{o}$ (298.15 K) = (133.0 ± 2.9) kJ·mol^−1^ $\ln\;\left(p/p_{ref}\right)=\frac{382.1}{R}-\frac{144576.8}{RT}-\frac{39.0}{R}\ln\frac{T}{298.15}$; *p*_ref_ = 1 Pa						
403.4	35.5	4417	4.159	0.109	128.8	235.5
386.5	12.9	8951	0.794	0.025	129.5	237.4
396.6	11.4	2756	2.208	0.060	129.1	236.4
406.8	8.2	14,260	5.602	0.165	128.7	235.0
366.7	11.3	75,182	0.084	0.007	130.3	238.9
376.5	10.2	20,355	0.279	0.012	129.9	238.7
2,2′,4,4′-tetra-hydroxy-benzophenone$:\;\Delta_{\mathrm{cr}}^{g}H_{m}^{o}$ (298.15 K) = (159.8 ± 3.1) kJ·mol^−1^ $\ln\;\left(p/p_{ref}\right)=\frac{404.6}{R}-\frac{172711.7}{RT}-\frac{43.2}{R}\ln\frac{T}{298.15}$; *p*_ref_ = 1 Pa						
431.8	8.31	17,366	0.251	0.011	154.1	249.6
436.6	11.87	14,993	0.414	0.015	153.9	249.4
441.9	20.99	16,139	0.676	0.022	153.6	248.7
448.2	13.67	18,965	1.236	0.036	153.4	248.2
448.9	14.96	5987	1.277	0.037	153.3	247.9
453.8	15.78	13,385	1.990	0.055	153.1	247.4
455.7	15.9	3250	2.433	0.066	153.0	247.5

^a^ Saturation temperature measured with the standard uncertainty (*u*(*T*) = 0.1 K). ^b^ Mass loss of the sample measured by weighing. ^c^ Vapour pressure at temperature *T,* calculated from the *m*. ^d^ Standard uncertainties were calculated with *u*(*p_i_*/Pa) = 0.005 + 0.025 (*p_i_*/Pa) for pressures below 5 Pa and with *u*(*p_i_*/Pa) = 0.025 + 0.025 (*p_i_*/Pa) for pressures from 5 to 3000 Pa. The standard uncertainties for *T*, *V*, *p*, *m*, are standard uncertainties with 0.683 confidence level. Uncertainty of the sublimation enthalpy *U*($\Delta_{\mathrm{cr}}^{g}H_{m}^{o}$) is the expanded uncertainty (0.95 level of confidence) calculated according to procedure described elsewhere [13,14]. Uncertainties include uncertainties from the experimental conditions and the fitting equation, vapour pressures, and uncertainties from adjustment of vaporisation enthalpies to the reference temperature *T* = 298.15 K.

**Table 3 molecules-27-08477-t003:** Compilation of data on molar heat capacities $C_{p,m}^{o}$ (cr or liq) and heat capacity differences at *T* = 298.15 K (in J·K^−1^·mol^−1^).

Compounds	$C_{p,m}^{o}\left(cr\right){\;}^{a}$	$-\Delta_{\mathrm{cr}}^{g}C_{p,m}^{o}{\;}^{b}$	$C_{p,m}^{o}\left(liq\right){\;}^{a}$	$-\Delta_{l}^{g}C_{p,m}^{o}{\;}^{b}$
2-hydroxy-benzophenone	225.4 [15]	34.6	366.4	105.8
3-hydroxy-benzophenone	233.7 [15]	35.8	366.4	105.8
4-hydroxy-benzophenone	226.0 [15]	34.7	366.4	105.8
2,4-dihydroxy-benzophenone	254.9	39.0	434.8	123.6
2,2′-dihydroxy-benzophenone	253.0 [16]	38.7	434.8	123.6
2-hydroxy-4-methoxy-benzophenone	286.6	43.8	422.8	120.5
2,2′-hydroxy-4-methoxy-benzophenone	301.0	45.9	491.2	138.3
2,2′,4,4′-tetrahydroxy-benzophenone	283.2	43.2	571.6	159.2
2,2′-dihydroxy-4,4′-dimethoxybenzophenone	347.0	52.8	547.6	153.0

^a^ Calculated by the group-contribution procedure developed by Chickos et al. [17]. ^b^ Calculated according to the procedure developed by Acree and Chickos [18].

**Table 8 molecules-27-08477-t008:** Thermochemical data for substituted benzophenones at *T* = 298.15 K (*p°* = 0.1 MPa) (in kJ·mol^−1^) ^a^.

Compound	$\Delta_{f}H_{m}^{o}(\mathrm{cr}/\mathrm{liq}){\;}^{a}$	$\Delta_{l,cr}^{g}H_{m}^{o}{\;}^{b}$	$\Delta_{f}H_{m}^{o}(g{)}_{\exp}$	$\Delta_{f}H_{m}^{o}(g){G_{4\;}}^{c}$ WBR	$\Delta_{f}H_{m}^{o}(g){G_{4\;}}^{d}$ AT	$\Delta_{f}H_{m}^{o}(g){G_{4\;}}^{e}$ (theor)
benzophenone		95.0 ± 0.4		49.3 ^f^	50.1	49.4 ± 0.8 ^g^
2-hydroxy-benzophenone (cr) [15]	−245.7 ± 3.8	97.9 ± 1.9	−147.8 ± 4.3	−150.5	−149.9	−150.2 ± 2.5
3-hydroxy-benzophenone (cr) [15]	−247.3 ± 4.0	131.7 ± 0.9	−115.6 ± 3.9	−125.1	−124.5	−124.8 ± 2.5
4-hydroxy-benzophenone(cr) [15]	−252.4 ± 3.3	130.3 ± 1.0	−122.1 ± 3.8	−129.4	−128.8	−129.1 ± 2.5
2-methoxy-benzophenone (liq)		87.4 ± 3.0		−100.3	−101.2	−100.8 ± 2.5
3-methoxy-benzophenone (liq)		88.7 ± 2.0		−106.4	−107.3	−106.9 ± 2.5
4-methoxy-benzophenone (liq)		90.2 ± 2.0		−107.6	−108.5	−108.1 ± 2.5
2,4-di-hydroxy-benzophenone (cr)	(−492.8 ± 1.9) [41]	134.5 ± 1.7	(−358.3 ± 2.5)	−333.6	−333.1	−333.4 ± 2.5
2-hydroxy-4-methoxy-benzophenone(cr) [25]	−414.2 ± 4.5	116.8 ± 1.2	−297.4 ± 4.7	−313.2	−314.2	−313.7 ± 2.5
2-hydroxy-2′-methoxy-benzophenone				−302.8	−303.8	−303.3 ± 2.5
2,2′-di-hydroxy-benzophenone (cr)		101.9 ± 1.3		−342.2	−342.2	−342.0 ± 2.5
2,2′-di-hydroxy-4-methoxy-benzophenone (cr)		118.7 ± 1.6		−505.2	−506.4	−505.8 ± 2.5
2,2′,4,4′-tetra-hydroxy-benzophenone (cr)		159.8 ± 3.1		−704.7	−704.7	−704.6 ± 2.5
2,2′-di-hydroxy-4,4′-dimethoxybenzophenone(cr)		139.4 ± 2.4		−667.0	−669.8	−668.4 ± 2.5

^a^ Uncertainties in this table are twice standard deviations. Values in brackets appear to be incorrect. ^b^ Taken from Table 4. ^c^ Calculated by the G4 method using reactions 9–14 and using experimental $\Delta_{f}H_{m}^{o}$(g)-values for the reaction participants. The expanded uncertainty assessed to be ±3.5 kJ·mol^−1^ [11]. ^d^ Calculated by the G4 method using atomisation reactions. The expanded uncertainty assessed to be ± 3.5 kJ·mol^−1^ [11]. ^e^ Weighted mean value. The uncertainties were taken as the weighing factor. ^f^ Calculated by the G4 method using reaction: benzophenone + butane = acetone+ 2× toluene and using experimental $\Delta_{f}H_{m}^{o}$(g)-values for the reaction participants [42]. ^g^ Results evaluated and recommended in our recent work [36].

**Table 9 molecules-27-08477-t009:** Parameters and pairwise nearest and non-nearest neighbour interactions of substituents on the “centrepieces” for calculation of thermodynamic properties of substituted benzenes and benzophenones at 298.15 K (in kJ·mol^−1^).

Centrepiece	$\Delta_{f}H_{m}^{o}(g)$	$\Delta_{l}^{g}H_{m}^{o}$
benzene [46]	82.9	33.9
benzophenone [36]	49.4	78.0
*Contributions* ^a^		
Δ*H*(H *→ CH_3_O*)	−153.2	12.7
Δ*H*(H *→ OH*)	−175.4	24.1
*Interactions* ^b^		
*ortho C = O*(*C_6_H_5_*) − *OH*	−24.5 ^c^	−21.5
*meta C = O*(*C_6_H_5_*) − *OH*	0.9 ^c^	9.1
*para C = O*(*C_6_H_5_*) − *OH*	−3.4 ^c^	12.0
*ortho C = O*(*C_6_H_5_*) − *CH_3_O*	3.5 ^c^	−2.9
*meta C = O*(*C_6_H_5_*) − *CH_3_O*	−2.6 ^c^	−1.6
*para C = O*(*C_6_H_5_*) − *CH_3_O*	−3.8 ^c^	1.5
*ortho OH* − *OH*	2.5	−11.4
*meta OH* − *OH*	−4.5	−3.7
*para OH* − *OH*	2.9	2.3
*ortho OH* − *CH_3_O*	−2.1	−9.3
*meta OH* − *CH_3_O*	0.1	4.1
*para OH* − *CH_3_O*	8.7	3.0
*ortho CH_3_O* − *CH_3_O*	17.5	5.2
*meta CH_3_O* − *CH_3_O*	−0.1	0.4
*para CH_3_O* − *CH_3_O*	7.3	2.3

^a^ The contributions were derived from the differences between the experimental enthalpy of the substituted benzene (or benzophenone) and the experimental enthalpy of benzene (or benzophenone) itself (see text). ^b^ The pairwise interactions between carbonyl and substituent were derived from the experimental enthalpies of substituted benzene (or benzophenone) and the corresponding enthalpy of the framework as shown in Figure 6. ^c^ From *H*_298_ values of the reverse reactions 9–10 participants calculated by using the G4 method.

**Table 10 molecules-27-08477-t010:** Strength of intra-molecular hydrogen bonding (intra-HB) in ortho-substituted benzophenones at *T* = 298.15 K (*p°* = 0.1 MPa, in kJ·mol^−1^).

Ortho-Substituted Benzophenones	$\Delta_{r}{H_{m}^{o}}_{\left(\mathrm{WBR}\right)}{\;}^{a}$	Δ*H*_WBR_ ^b^	Δ*H*_conf_ ^c^	Δ ^d^
2-hydroxy-benzophenone	−24.5	−24.5	−35.7	11.2
2,4-di-hydroxy-benzophenone	−32.2	−24.3	−43.6	19.3
2-hydroxy-4-methoxy-benzophenone	−34.0	−30.3	−43.9	13.6
2,2’-di-hydroxy-benzophenone	−40.8	−40.8 (−20.4) ^e^	−68.6	27.8
2,2’-di-hydroxy-4-methoxy-benzophenone	−50.7	−47.0 (−23.5) ^e^	−75.8	24.4
2,2’,4,4’-tetra-hydroxy-benzophenone	−52.5	−36.7 (−18.4) ^e^	−74.9	38.2
2,2’-di-hydroxy-4,4’-dimethoxybenzophenone	−59.2	−51.8 (−25.9) ^e^	−83.4	31.6
2-hydroxy-2’-methoxy-benzophenone	−23.6	−27.1	−41.3	14.2
2-methoxy-benzophenone	3.5	-	-	-

^a^ Reaction enthalpies calculated using the G4 method and the WBR reactions 9–14 (in reverse notation). ^b^ Intra-HB strength derived from the enthalpies of WBR reactions 9–14 (column 2) after corrections for pairwise interactions of substituents were performed as shown in Appendix A. ^c^ Intra-HB strength is calculated using the G3MP2 and the “HB and Out” method. ^d^ Difference between columns 3 and 4. ^e^ Intra-HB strength calculated per single bond.

## Data Availability

Not applicable.

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
