# Peer review of "Nearest-Neighbour and Non-Nearest-Neighbour Non-Covalent Interactions between Substituents in the Aromatic Systems: Experimental and Theoretical Investigation of Functionally Substituted Benzophenones"

_molecules, 2022, doi:10.3390/molecules27238477_

Round 1
Reviewer 1 Report
The submitted manuscript present the results of combined experimental and theoretical study, described in the comprehensive way. The work is at the high level, however I have some questions and issues that should be revised.
The title is too long. The first sentence “Nearest-neighbour and non-nearest-neighbour non-covalent interactions between substituents in the aromatic systems.” should be removed.
I really appreciate the application of G4, it should be a golden standard in such studies, unfortunately some people still use B3LYP in similar cases… Well done!
Line 84, the DSC experimental part must be described in more detailed way, i.e. what type of calorimeter has been used and how the analysis has been performed.
Table 4, why the value for 2,4-di-hydroxy-benzophenone from the [24] has been excluded, although it is within the uncertainty range with the one from Table 7? The same applies to the 2,2 ́,4,4 ́-tetrahydroxy-benzophenone.
Line 310, I am a little worried about the basis set being to small to provide accurate results, even if they are only used to compare the selected conformers.
Figure 3, why haven’t you considered the possibility of methyl group rotation in the derivatives with the methoxy group? This can also have a considerable impact on the relative energy of the conformers.
Line 488, what was the refcode of this mentioned structure?
Reviewer 2 Report
Manuscript ID: molecules-2036463
Comments to authors:
The authors report an experimental and theoretical investigation on benzophenones substituted with hydroxyl and methoxyl groups, in order to try to obtain reliable group-additivity predictive schemes. The work described seems to be performed carefully. In my opinion, the article will be suitable for publication in Journal Molecules, after minor revision to improve some aspects.
The authors should consider the following points:
1) Page 2, line 52 – the sentence has a misprint.
2) Table 8 – last column, except the first value, the remain uncertainties are correct? For G4, the expanded uncertainty assessed is 3.5 kJ/mol.
3) Table 9:
a. The contributions presented are only for benzene. The footnote “a” should be rewritten.
b. According to the values in Table S1, the contribution of methoxy group for the enthalpy of vaporization is not correct. Please, confirm the values and the calculation.
4) Figure 6, the legend has a misprint.
5) Page 20, lines 418-420, the explanation about how to get the theoretical framework is confused. As suggestion, it would be better to clarify that the contribution delH(H-OH) will be add to the benzophenone – centerpiece. Also, an additional information in parenthesis (See examples from figures 7 and 8) it could help.
6) Page 21, line 423 – What the authors mean with the arithmetic difference? For the example of the text, the authors used the framework (+126.0 kJ/mol) and instead of the experimental value, the authors used the WBR value obtained with G4 method.
Footnote “c” – values used are only for the WBR or also with the AT?
Please, clarify that, if necessary, in the text of the manuscript.
7) Page 24, line 510 has a misprint.
8) Figure 14, the size should be reduced.
Round 2
Reviewer 1 Report
The Authors have made the requested correction. This work is now complete and can be published as it is.